# Intelligent Diagnostics of Radial Internal Clearance in Ball Bearings with Machine Learning Methods

**DOI:** 10.3390/s23135875

**Published:** 2023-06-25

**Authors:** Bartłomiej Ambrożkiewicz, Arkadiusz Syta, Anthimos Georgiadis, Alexander Gassner, Grzegorz Litak, Nicolas Meier

**Affiliations:** 1Department of Automation, Faculty of Mechanical Engineering, Lublin University of Technology, Nadbystrzycka 36, 20-618 Lublin, Poland; g.litak@pollub.pl; 2Institute of Production Techniques and Systems, Leuphana University of Lüneburg, Universitätsallee 1, 21335 Lüneburg, Germany; georgiadis@leuphana.de (A.G.); lg076279@stud.leuphana.de (A.G.); nmeier@leuphana.de (N.M.); 3Department of Computerization and Robotization of Production, Faculty of Mechanical Engineering, Lublin University of Technology, Nadbystrzycka 36, 20-618 Lublin, Poland; a.syta@pollub.pl

**Keywords:** ball bearings, radial internal clearance, time-series analysis, machine learning

## Abstract

This article classifies the dynamic response of rolling bearings in terms of radial internal clearance values. The value of the radial internal clearance in rolling-element bearings cannot be described in a deterministic manner, which shows the challenge of its detection through the analysis of the bearing’s dynamics. In this article, we show the original approach to its intelligent detection through the analysis of short-time intervals and the calculation of chosen indicators, which can be assigned to the specific clearance class. The tests were carried out on a set of 10 brand new bearings of the same type (double row self-aligning ball bearing NTN 2309SK) with different radial internal clearances corresponding to individual classes of the ISO-1132 standard. The classification was carried out based on the time series of vibrations recorded by the accelerometer and then digitally processed. Window statistical indicators widely used in the diagnosis of rolling bearings, which served as features for the machine learning models, were calculated. The accuracy of the classification turned out to be unsatisfactory; therefore, it was decided to use a more advanced method of time series processing, which allows for the extraction of subsequent dominant frequencies into experimental modes (Variational Mode Decomposition (VMD)). Applying the same statistical indicators to the modes allowed for an increase in classification accuracy to over 90%.

## 1. Introduction

Rolling bearings are the basic element of transferring rotational motion in many mechanical systems, and thus they are exposed to many changes, causing a decrease in their life as well as reducing work safety. One such change is also a change in the value of the radial internal clearance. Radial internal clearance in ball bearings is important because it helps to ensure that the bearing runs smoothly and without excessive vibration or noise. It also allows for proper lubrication, which reduces wear, tear, and fatigue on the components of the bearing. Radial internal clearance ensures that there is enough space between each ball and its inner race so that they can move freely within their respective shape errors. Without adequate radial internal clearance, the balls will rub against each other as they rotate, resulting in increased friction and heat buildup, which can cause premature failure of the rolling-element bearing. Its change also causes a change in the physical effects of the bearings and, consequently, a qualitative and quantitative change in the non-linear system’s dynamic response [1,2,3,4].

To present the importance of radial internal clearance on the performance of rolling-element bearings, it is worth referring to the historical background when both the mathematical model and experimental approach were studied. Tiwari et al. [5] in an experiment studied the influence of the radial internal clearance on the dynamic response of the rotor, presented in the form of orbit plots, cascade plots, and frequency plots. He proved the influence of the radial internal clearance on the unbalance of the whole rotor, showing both periodic, sub-harmonic, and chaotic responses of the system depending on the clearance value and the rotating velocity of the shaft. The first advanced mathematical model of rolling-element bearing was discussed by Changqing et al. [6], in which the influence of the following features, such as the radial internal clearance, shape errors, and high-speed effect, was related to the centrifugal force and gyroscopic moment. In the mentioned research, it was shown that clearance, axial preload, and radial force play a significant role in affecting system stability. Another mathematical modeling approach was discussed by Upadhyay et al. [7]. The results obtained showed the appearance of instability and chaos in the dynamic response as the speed of the rotor-bearing system and value of radial internal clearance changed. There are some features regarding the variability of the radial internal clearance, which, without doubt, the contamination and thermal effects are. In the thermographic inspection, Miskovic et al. [8] found a dependence between the amount of contamination and its effect on the operating clearance value during rotation [9]. The recent trend is focused on the automatic diagnosis of radial internal clearance by studying the dynamics of rolling-element bearings. Xu et al. [10,11] looked for accurate indicators allowing an automatic diagnosis of radial internal clearance in wind turbines. He showed that such indicators as modulation signal bispectrum-sideband (MSB-SE), root mean square (RMS), and spectral centroid can be used for clearance diagnostics in ball bearings, showing their high potential in studying the dynamical response obtained with the mathematical model. The recent trend in studying the impact of radial internal clearance on bearing dynamics gives motivation towards the automatic diagnostics of radial internal clearance value, which can be achieved with finding accurate diagnostic indicators and the application of nonlinear time-series analysis combined with artificial intelligence methods (AI) [12,13]. The Machine Learning (ML) methods are so useful and provide such high accuracy in classification that they have already been applied in the following areas of research: maritime installations [14], smart homes [15], cognitive behavior [16], fault diagnostics [17], analysis of climate changes [18], environmental protection [19], and many more.

Rolling-element bearings have strongly nonlinear characteristics due to multiple factors such as shape errors [20], defects [21], friction [22], or contamination [23]. In the case of evident fault detection, the Fast Fourier Transform (FFT) easily describes the characteristic frequencies corresponding to the faults of the specific element of a rolling-element bearing [24,25,26]. In the diagnostics of bearings, the following trends can be specified:Predictive maintenance that became significant trend in the diagnostics of rolling-element bearings, allowing for early detection of faults and potential failures [27,28].The integration of advanced sensor technologies, such as vibration analysis, temperature monitoring, and acoustic emission analysis, has enabled more accurate and comprehensive diagnostics of rolling-element bearings [29,30].The use of artificial intelligence and machine learning algorithms has gained prominence, as they can analyze large volumes of sensor data and identify patterns indicative of bearing degradation or impending failure [31].Non-destructive testing methods, such as ultrasound and infrared thermography, are increasingly being employed for bearing diagnostics, providing valuable insights into internal defects and anomalies [32,33].Remote monitoring and connectivity solutions have emerged as a significant trend, enabling real-time data acquisition from bearings installed in remote or inaccessible locations, thereby facilitating proactive maintenance and reducing downtime [34,35].

Referring to the considered problem, which is the diagnostics of radial internal clearance under different operating conditions, more sophisticated methods have to be used due to its variability and fluctuations in time. In the previous paper, the accuracy of some recurrence quantificators was proven [4], but other statistical indicators can also be used for the preparation of the window analysis [36,37,38]. The current study refers to the previously discussed issue; however, the window analysis is based on nine statistical indicators. The motivation is the automatic identification of the radial internal clearance in the self-aligning ball bearing with a conical bore, in which it is possible to control its value in a wide range [39]. For automated prognostics of radial clearance values even with different operational velocities, the experiment is automated with Machine Learning algorithms [40,41,42]. Basic window analysis does not give an accurate ML model because first signals are decomposed with Variational Mode Decomposition (VMD) [43,44], and then the procedure is repeated, increasing the accuracy of the ML model to over 90%. The novelty of this paper in reference to the previous publications is the intelligent and subsequently automatic radial internal clearance classification by its classes. The challenge of the considered problem is that the clearance level cannot be described in a deterministic way and is a strictly nonlinear factor in the operation of a rolling-element bearing.

After the introduction to this paper, the remainder of this paper is as follows. In Section 2, the test rig and the experimental procedure are discussed. Section 3 refers to the data processing and applied methods. Additionally, a mathematical description of applied methods is presented. In Section 4, the results of VMD and machine learning are obtained, and a discussion of them is given. Conclusions summarize the paper.

## 2. Experimental Setup and Procedure

The experimental setup used in the experiment consists of two test rigs, i.e., an automated setup used for the measurement of radial internal clearance in rolling-element bearings and the test rig used for the dynamical test in which the rotational velocity is under control. The experiment was conducted in the laboratory of the Institute of Product and Process Innovation at Leuphana University of Lüneburg (Germany). The contribution of our research group can be divided into three subgroups dealing with the preparation of experiments and data acquisition, signal processing and application of Machine Learning methods, and supervision over the conducted research.

### 2.1. The Automated Setup for Measuring Radial Internal Clearance

For the purpose of automating the process of radial internal clearance measurement, a novel test rig has been built and patented [39]. In our experiment, we are focused on tapered bore bearings, which refers to their conical shape and the possibility of changing the internal clearance with the help of axial force in its wide range. Moreover, the test rig is digitalized, which increases precision and accuracy; however, the measurement process of clearance is still followed according to the international standard ISO-1132. The dial gauges in the test rig are used to measure the value of clearance, the distance of shifting the bearing onto the adapter sleeve with the conical diameter, and the influence of the test force on the displacement on the shaft (Figure 1).

### 2.2. Test Rig for Dynamical Tests

After setting the specific value of radial internal clearance, the ball bearing with the shaft was mounted in the plummer block and interconnected with a coupling to the electric motor. The test rig allows testing of the bearing up to 3000 rpm, corresponding to 50 Hz, while the test is conducted with the velocity step every 10 Hz. To the plummer block, two accelerometers are attached, and one of them is used for collecting the acceleration data (vertical direction). The sampling frequency during the test is equal to 1562.5 [Hz], which corresponds to a sampling time of 0.64 [ms]. The type of bearing used in the experiment is a self-aligning double-row ball bearing with a conical bore. The initial clearance in most of the studied bearings is more than 40μm, and owing to the conical bore, it is possible to decrease its value to around 8μm. The details of the used test rig (Figure 2) and the applied experimental procedure are described in detail in the following paper [4]. The elements creating the experimental test rig are specified in Table 1, while the features of the studied bearing are specified in Table 2, and its characteristic frequencies are in Table 3.

It was impossible to set the same value of radial internal clearance in each of the tested bearings; this is why similar cases were induced in each of them. The values of radial clearances were ranged according to the specific clearance classes, which are specified in Table 4. The abbreviations in Table 4 refer to the names of clearance classes; i.e., C2L corresponds to lower values of radial clearance in the C2 class; CN refers to the range of normal clearance, etc. The classes are specified according to the ISO-1132 standard.

## 3. Data Processing and Applied Methods

Over the years, a variety of digital signal processing (DSP) methods have been applied to the diagnostics of rotational systems. These methods include frequency-domain techniques such as the Fast Fourier Transform (FFT), Variational Mode Decomposition (VMD), Spectral Kurtosis (SK), and others. These techniques are supported by analysis performed in the time domain, which allows us to observe and detect dynamic changes present in the system over a specific period of time. 

In this study, we used data from 10 identical new bearings tested with different rotational speeds and different internal clearance. In the previous case study research, we have proved with nonlinear methods that different values of the internal clearance (different classes of the ISO-1132 standard) result in a different dynamic response of the system. Instead of using group statistics, machine learning models were used to classify individual classes due to different values of internal clearance. Time series of vibrations registered in the y direction were used (vibrations measured in this axis turned out to carry the most information about the state of the system). The exemplary acceleration time-series and its power spectrum is presented in Figure 3. The methodology for preparing training and test data can be divided into two stages. The first is raw data analysis without advanced digital signal processing techniques. The second is the decomposition (VMD) of each series into periodic components centered around the dominant frequency (IMFs) and treated as new data (Figure 4). In both cases, statistical indicators were calculated (in non-overlapping windows with a length of 2000 points each), widely used in the diagnostics of rolling bearings [45]: Mean, Median, Variance, Kurtosis, Skewness, Peak to Peak, Crest Factor, FM4, and NA4. The ratio of data for training and validation is equal to 75% (6300 data points) to 25% (2100 data points) which corresponds in total 8400 data points. In Figure 5, the flowchart of data processing is presented, choosing the analysis with and without VMD and next the application of ML methods.

Applying statistical condition indicators (CI) to time series data obtained from experiments has proven to be an accurate method for diagnosing different types of damage or transient states occurring in gears, bearings, shafts, and other rotational systems. These indicators can be divided into different domains, including the time domain, frequency domain, and time–frequency domain. In our experiment, we focus on a subset of these indicators that we believe will have the most significant impact on identifying bearing responses with different RICs. By selecting a subset of statistical condition indicators, we can focus on the most relevant features for identifying bearing responses with different RICs in our experiment. This can improve the accuracy and effectiveness of our diagnostic approach. For most of them, their mathematical description is well known, while two of them require better explanation.

FM4 describes how peaked or flat the amplitude of the difference signal is. The FM4 indicator is normalized by the square of the variance, which helps to reduce the impact of changes in the overall vibration level of the machine.(1)$$\mathrm{FM}4=\frac{N\sum_{n=1}^{N}{\left(d_{i}-\bar{d}\right)}^{4}}{{\left[\sum_{n=1}^{N}{\left(d_{n}-\bar{d}\right)}^{2}\right]}^{2}}$$
where *d* is the difference signal, $\bar{d}$ is the mean value of difference signal, *N* is the total number of interpolated data points per reading, and *i* is the interpolated data point number.

2.NA4 can provide an indication of the onset of damage as well as the magnitude of the damage as it progresses. The mentioned indicator is a measure of the variation in the amplitude of the vibration signal from a rotating machine over a certain time period. (2)$$\mathrm{NA}4=\frac{N\sum_{n=1}^{N}{\left(r_{i}-\bar{r}\right)}^{4}}{{\left[\sum_{n=1}^{N}{\left(r_{n}-\bar{r}\right)}^{2}\right]}^{2}}$$
where r is the residual signal, $\bar{r}$ is the mean value of residual signal, *N* is the total number of interpolated data points per reading, and *n* is the interpolated data point number.

Due to the complexity of the underlying non-linear system and its sensitivity initial conditions, it was decided to decompose each of the signals into modes (IMFs) using the Variational Mode Decomposition (VMD) algorithm. Variational Mode Decomposition (VMD) is a data-driven signal processing technique that decomposes signals into meaningful components [46]. It is based on the variational mode analysis method, which has been used to analyze various types of nonlinear systems. The VMD algorithm works by finding the oscillatory components that best represent the signal using a variational principle. VMD uses an iterative procedure to identify meaningful components from a given signal by minimizing an energy functional associated with each IMF. The obtained signals cover the original signal’s frequency spectrum with different frequency components. One advantage of VMD is that it can be used for non-stationary signals with time-varying frequencies, which is common in many real-world applications. Choosing the proper number of intrinsic mode functions (IMFs) typically involves considering the complexity and characteristics of the underlying signal. A general guideline is to select a sufficient number of IMFs to capture the essential features of the signal without introducing excessive noise or redundancy. Balancing interpretability and fidelity to the original signal are crucial when determining the optimal number of IMFs for a specific application, in our case the analysis of acceleration time-series by different value of radial internal clearance [47,48]. This allows for the extraction of multiple underlying modes in a single pass, resulting in more accurate and efficient feature extraction than traditional methods such as Fourier analysis or wavelet transform [49]. It is similar to the widely used Empirical Mode Decomposition (EMD) algorithm [50] and its variations but can be more robust to frequency mixing between modes. The input signal is defined as a sum of amplitude and frequency modulated signals:(3)$$x\left(t\right)=\sum_{n=1}^{N}v_{n}\left(t\right)=\sum_{n=1}^{N}A_{n}\left(t\right)\cos\left(\varphi_{n}\left(t\right)\right)$$
where A_n_(t) is the amplitude and φn(t) is the phase. Each IMF is described by slowly varying and positive envelops and non-decreasing instantaneous frequency concentrated around central frequency. The method finds both the modes amplitudes and corresponding central frequencies (in the same time) by minimizing the constrained variational problem:(4)$$\underset{\left\{v_{n},f_{n}\right\}}{\min}\left\{\underset{n}{\sum }\|\partial_{t}\left[\left(\delta \left(t\right)+\frac{j}{\pi t}\right){*v}_{n}\left(t\right)\right]e^{-{\mathrm{jf}}_{n}t}{\|}_{2}^{2}\right\}$$
where {v_n_} = {v_1_, v_2_, …, v_N_} denotes the set of all modes, {f_n_} = {f_1_, f_2_, …, f_N_} denotes the set of central frequencies, δ is the Dirac function, ‖·‖ is the L_2_ norm, and ∗ is the convolution operator. The term $\left(\delta \left(t\right)+\frac{j}{\pi t}\right)*v_{n}\left(t\right)$ defines the analytical signal, and the term *e*^−*jfnt*^ defines the frequency spectrum of the baseband. Note that the choice of the number of modes is critical. In this paper, the number of mods has been fixed at 10.

## 4. Results and Discussion

The comparison of the model’s accuracy at both the training and validation stages was one using one of the automatic machine learning package, in this case PyCaret [51]. PyCaret streamlines the workflow of traditional machine learning and automates many tasks such as feature engineering, model selection, and hyperparameter tuning. As mentioned earlier, in the first stage of the research, the window statistics were applied to the raw data generating the features within the data set. Table 5 shows the four models with the highest learning accuracy in the training process. The best accuracy has been obtained for models based on decision trees.

Light Gradient Boosting (LGB) is a machine learning algorithm that uses gradient boosting to make predictions. It is an efficient and versatile algorithm, which makes it suitable for many applications such as ranking, classification, and regression problems. LGB works by combining weak learners in a series of iterations to create an ensemble model with the highest accuracy possible [52,53].

Extreme Gradient Boosting (XGBoost) is another machine learning algorithm based on gradient boosting that has become popular over recent years due to its superior performance compared with traditional methods such as Random Forests or Support Vector Machines. The algorithm works by building an ensemble of decision trees with high accuracy. It uses regularization techniques such as shrinkage and column subsampling for better performance over traditional gradient boosting methods like GBM (Gradient Boosted Machines) [54,55].

Extra trees (ET) are an ensemble method, combining multiple decision trees that have been built using a random subset of the features. This allows for more accurate predictions and greater generalization than single decision trees [56,57].

Random forests (RF) are also an ensemble method, but they use a different approach: each tree is grown using only a randomly selected subset of the data points in the training set and only a randomly selected subset of the features when making splits at each node. The result is an even more robust model than extra trees, with better accuracy and less overfitting. Detailed descriptions of the models as well as examples of implementation can be found in [58,59].

In all cases, default model parameters have been taken, without their optimization, that are contained in scikit-learn package. Scikit-learn is a popular Python library for machine learning that provides a wide range of tools and algorithms for tasks such as classification, regression, clustering, and dimensionality reduction. It is widely used for its simplicity, versatility, and integration with other scientific computing libraries in the Python ecosystem.

Comparing the average accuracy of the classification at the stage of learning the models, you can notice its highest value for the lowest speeds (10 Hz and 20 Hz) and its clear decrease for higher speeds (above 20 Hz). The obtained result is consistent with the fact that damage (in this case, a change in the dynamics of the state of the system) in rotary machines is usually most visible at the lowest rotational speed. On the other hand, the classification accuracy values indicate a possible improvement of the score especially for higher speeds. In the next stage of the research, it was decided to increase the number of features by decomposing the signal into intrinsic modes modes using the VMD algorithm [60,61]. Then, each of the new signals was subjected to the same window analysis as in the first phase. Table 6 shows the percentage results of the learning stage of the models used earlier (Table 5). When comparing the model training results on data with more features (Table 7), you can see an increase in accuracy by up to 30% for speed greater than 30 Hz and about a 10% increase for speed lower than 30 Hz. It is also worth mentioning that the average learning time of the model did not exceed 5 s, amounting to 0.9 s (LGB), 4.29 s (XGBoost), 0.12 s (ET), and 0.32 s (RF), respectively.

In the next phase of classification, the model with the highest average accuracy was selected, i.e., Light Gradient Boosting, which performed the classification on a previously unseen test data. The differences in the accuracy of the Light Gradient Boosting (LGB) classifier between the learning process (Table 6) and testing (Table 7) are small, which indicates no overfitting effect of the model. Average accuracy is a reliable indicator of the model’s performance, but it does not reflect the accuracy of classification between different classes. This can be analyzed using the confusion matrix, which allows the visualization of the performance of an algorithm [62].

A confusion matrix is a table that is often used to describe the performance of a classification model (or “classifier”) on a set of data for which the true values are known. Each column of the matrix represents the instances in a predicted class while each row represents the instances in an actual class (or vice versa). The name stems from the fact that it makes it easy to see if the system is confusing two classes (i.e., commonly mislabeling one as another). One vs. the rest of the classification results is shown in the graph (Figure 6). When analyzing the confusion matrices, it can be noticed that class C2 is most accurately predicted regardless of the rotational speed. The largest false prediction occurs for the middle classes (C2L, CN, and C3) and reaches up to 8% for the speeds of 30 Hz and 50 Hz.

The analysis of features in terms of their importance on prediction can also provide relevant information. In the case of the LGB model, the importance of individual features was determined as the number of splits of data across all trees. Figure 7 presents the top ten features of importance of the LGB classifier. The influence of individual features on the effectiveness of the classification shows a certain repeatability (Figure 7). Firstly, in all cases one feature is dominant, i.e., the value of FM4 for the last component (Imf_10), which is concentrated around the lowest dominant frequency. Secondly, the first ten top features with the highest significance in most cases were determined based on kurtosis (Fm4, Na4, Skewness), and in a few cases it is variance (Var). As predicted, low frequency signals (Imf_i where i ∈ {10, 9, 8, 7}) appear to be the most important, but there are also signals with the highest frequencies (Imf_i where i ∈ {1, 3}). Another approach to the research on reliability of results obtained is the cross-validation (CV) [63,64,65], which will be considered in the next research.

## 5. Conclusions

This article presents the application of machine learning methods to classify the internal clearance of rolling bearings. The vibration signals of 10 new bearings with different initial clearances recorded on the experimental stand were subjected to feature extraction using window statistics widely used in bearing diagnostics. Each class corresponds to a different range of internal clearance according to the ISO-1132 standard. In the first approach, the determined values of statistical indicators were used as input data for various classification models. At the learning stage, it turned out that the best classification models were based on the structure of decision trees (Table 5). This was especially noticeable at low rotational speeds, with a classification accuracy of greater than 83% for 10 Hz and about 80% for 20 Hz. Higher speeds showed a decrease in accuracy, which ranged between 55% and 58%. In order to improve accuracy, the set of features was enlarged 10 times, determining statistical indicators based on the extracted experimental components focused around 10 dominant frequencies. The new training data set has improved the classification accuracy depending on speed (Table 6): for low speeds (10 Hz and 20 Hz) by more than 10 percent to almost 94% and for higher speeds (30 Hz, 40 Hz, and 50%) by more than 30 percent to almost 89%. At the testing stage, the best-performing classifier was selected, i.e., the Light Gradient Boosting model, which showed accuracy in the range of 86% to 94% (Table 7). Detailed insight into the classification between the individual classes using the confusion matrix (Figure 6) showed the highest number of correct classifications (up to 96%) for the C2 class in all cases. On the other hand, the most incorrect classifications (up to 8% percent) were obtained for the middle classes C2L, CN, and C3. Moreover, it turned out that low-frequency signals carry the most information, although high-frequency components are also important (Figure 7). Summing up the above, using only the acceleration vibration signals, it was possible to classify the internal clearance in rolling bearings at 94%. Such a high efficiency was achieved by the Light Gradient Boosting model, although the other models based on tree architecture were not much behind it. The obtained results may be helpful in the classification of the dynamic condition of the bearing, which may change depending on the size of the clearance. It should also be noted that the resulting inaccuracies in the classification may be related to slight changes in the initial conditions associated with each installation of a new bearing on the test bench. The next step of the research will be a more strict analysis of the number of chosen IMFs and the analysis and cross-validation of the data applied to testing and validation.

## Figures and Tables

**Figure 1 sensors-23-05875-f001:**
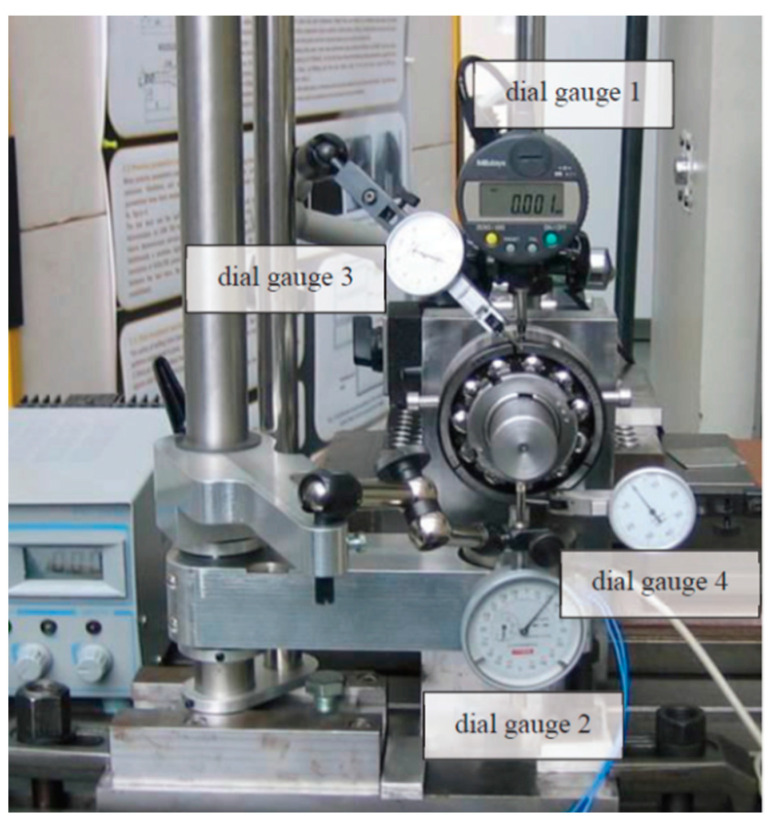
The automated setup for measuring bearing clearance [39].

**Figure 2 sensors-23-05875-f002:**
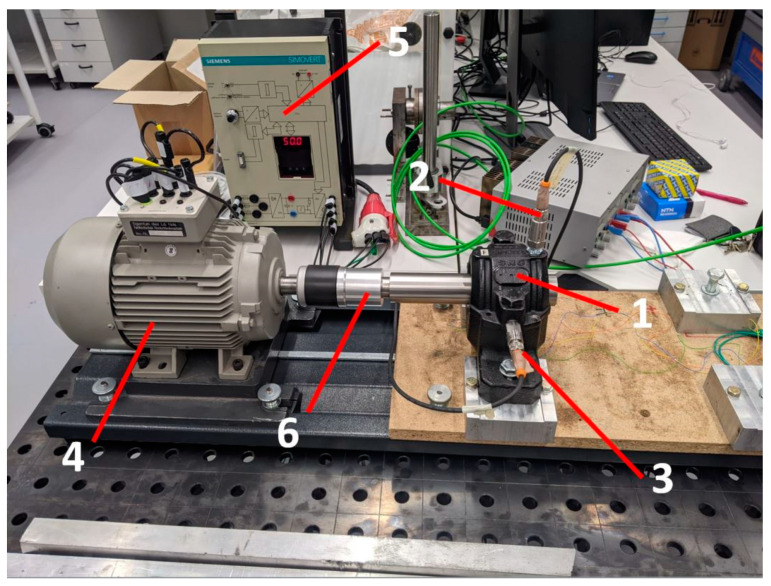
Experimental setup: 1—plummer block with installed ball bearing, 2—vertical accelerometer, 3—horizontal accelerometer, 4—3-phase motor, 5—inverter, 6—coupling [4].

**Figure 3 sensors-23-05875-f003:**
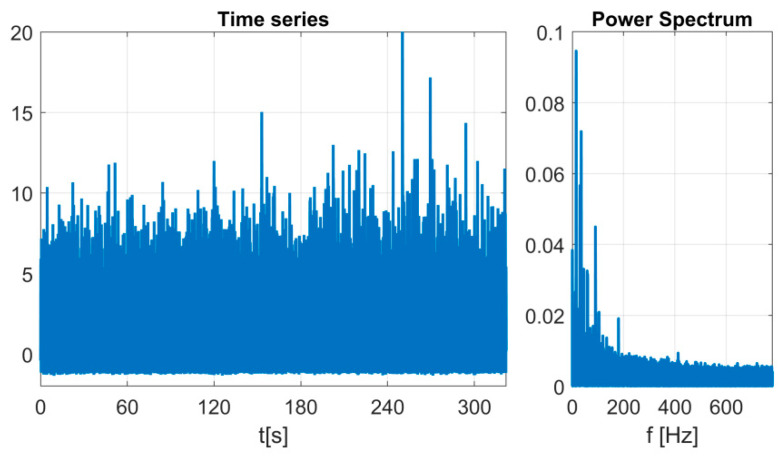
Time series of acceleration in the y direction for the first bearing for internal clearance 25 [μm] and rotational speed 30 [Hz] (**left**) and corresponding power spectrum (**right**).

**Figure 4 sensors-23-05875-f004:**
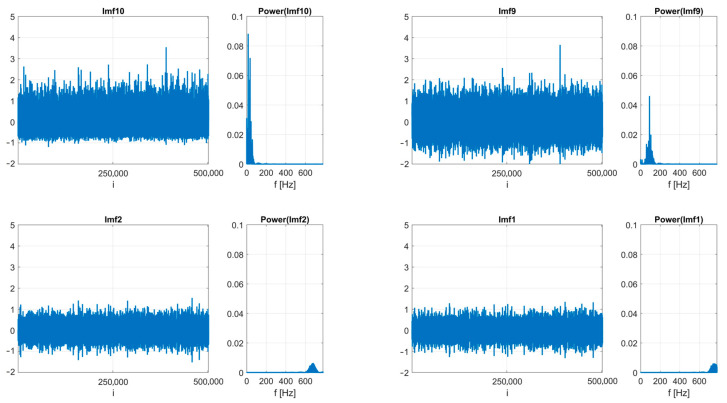
Time series of 2 lowest and 2 highest frequency components with their power spectra of time series presented in Figure 3.

**Figure 5 sensors-23-05875-f005:**
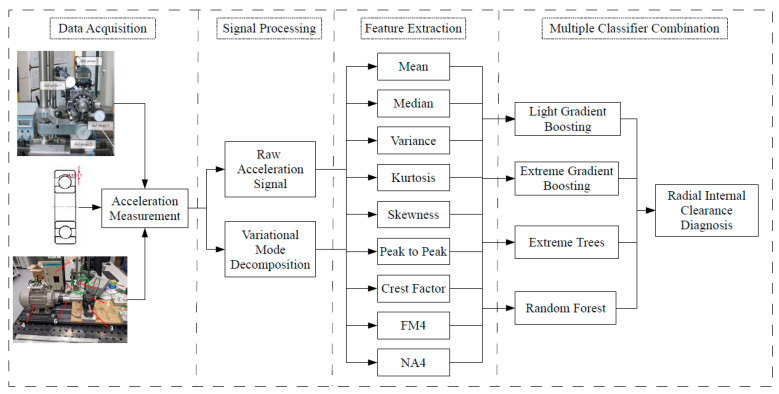
Flowchart of the data processing.

**Figure 6 sensors-23-05875-f006:**
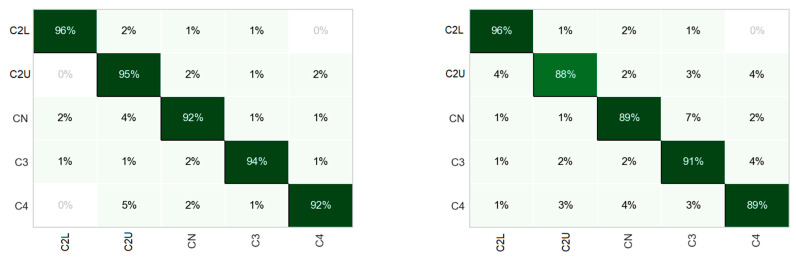
Confusion matrices for LGB model classification with increasing speed from 10 Hz to 50 Hz (from left top to the bottom).

**Figure 7 sensors-23-05875-f007:**
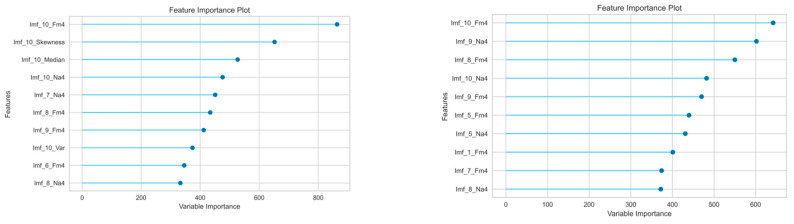
Top ten features of the LGB model classification with increasing speed from 10 Hz to 50 Hz (from left top).

**Table 1 sensors-23-05875-t001:** Equipment for experimental setup.

Component	Type
Ball Bearing	NTN 2309SK
Adapter Sleeve	NTN H2309
Locating Ring	NTN FR 100 × 4
Bearing Housings	NTN SNC 511-609
Vertical Accelerometer	IFM VSP001 (Piezo)
Horizontal Accelerometer	IFM VSA001 (MEMS)
Diagnostic Box	IFM VSE100
3-Phase Motor	Siemens 1LA5090-4AA60-Z
Inverter	Siemens SIMOVERT 6SE2103
Eddy Current Sensor	DT 3300

**Table 2 sensors-23-05875-t002:** Relevant dimensions of tested bearing.

Component	Type
Ball Diameter *b_d_* [mm]	15.870
Pitch Diameter *d_p_* [mm]	71.810
Pressure Angle *β* [°]	15.52
Number of Rolling Elements *b_n_*	26 (13 per row)
Number of Bearings tested in the Experiment	10

**Table 3 sensors-23-05875-t003:** Characteristic frequencies of tested bearing.

Characteristic Frequency	Value
Fundamental Train Frequency (FTF) [Hz]	0.394
Ball Spin Frequency (BSF) [Hz]	2.160
Ball Pass Frequency Inner Ring (BSFI) [Hz]	7.884
Ball Pass Frequency Outer Ring (BSFO) [Hz]	5.116

**Table 4 sensors-23-05875-t004:** Specified clearance classes in the experiment.

Marking	Range of Clearance
C2L	8–11 [μm]
C2U	14–16 [μm]
CN	18–24 [μm]
C3	25–36 [μm]
C4	38–48 [μm]

**Table 5 sensors-23-05875-t005:** Percentage comparison of the overall accuracy of the raw data model learning process.

Velocity\ Classifier	10 [Hz]	20 [Hz]	30 [Hz]	40 [Hz]	50 [Hz]
Light Gradient Boosting	83.6%	79.7%	56.7%	57.7%	57.4%
Extreme Gradient Boosting	83.5%	79.7%	55.8%	57.3%	56.6%
Extra Trees	83.9%	80.2%	55.6%	56.8%	57.1%
Random Forest	83.5%	79.7%	55.9%	57.1%	57.7%

**Table 6 sensors-23-05875-t006:** Percentage comparison of the overall accuracy of the model on training data.

Velocity\ Classifier	10 [Hz]	20 [Hz]	30 [Hz]	40 [Hz]	50 [Hz]
Light Gradient Boosting	93.9%	91.2%	85.0%	88.4%	88.6%
Extreme Gradient Boosting	93.6%	90.1%	84.1%	88.3%	88.1%
Extra Trees	91.7%	87.6%	81.3%	86.3%	86.8%
Random Forest	91.6%	90.1%	83.2%	87.1%	87.9%

**Table 7 sensors-23-05875-t007:** Percentage comparison of the overall accuracy of the model on training data.

Velocity\ Classifier	10 [Hz]	20 [Hz]	30 [Hz]	40 [Hz]	50 [Hz]
Light Gradient Boosting	94.3%	91.1%	86.2%	89.9%	89.4%

## Data Availability

Data are available on request due to privacy restrictions.

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
