# Peer review of "Intelligent Diagnostics of Radial Internal Clearance in Ball Bearings with Machine Learning Methods"

_sensors, 2023, doi:10.3390/s23135875_

Round 1

Reviewer 1 Report

This article classifies the dynamic response of rolling bearings in terms of radial internal clearance values. The tests were carried out on a set of brand new bearings of the same type (double row self-aligning ball bearing NTN 2309SK) with different radial internal clearance corresponding to individual classes of the ISO-1132 standard.

(1)The abstract should be improved. Your point is your own work that should be further highlighted.

(2)The parameters in expressions are given and explained.

(3) The method in the context of the proposed work should be written in detail.

(4)In Section 2, the authors should provide the describing of contributions.

(5) In Section 4, how to determine these parameter values for used methods.

(6) What is the time complexity of proposing a method? The author should provide.

(7) The literature review is poor in this paper. I hope that the authors can add some new references in order to improve the reviews.

https://doi.org/10.1109/TR.2022.3180273; http://dx.doi.org/10.1109/TCSS.2022.3152091; http://doi.org/10.3390/app13095706; http://dx.doi.org/10.1016/j.marstruc.2022.103338 and so on.

(8) There are some grammatical errors seen in the paper. Check carefully for a few clerical errors and formatting issues.

 There are some grammatical errors seen in the paper. Check carefully for a few clerical errors and formatting issues.

Author Response

Lublin, 17.06.2023

Response to Reviewer no. 1

Dear Reviewer,

We appreciate the constructive suggestions enclosed in your review. All the authors of this manuscript are grateful for Reviewer’s encouraging approach, as well as constructive criticism that is intended to improve our work. We carefully considered the comments
and herein we explain how we revised the manuscript based on your recommendations.

It is our belief that the manuscript is substantially improved after making the suggested edits. We want to express our appreciation for taking the time and effort necessary to provide guidance.

Yours sincerely

Bartłomiej Ambrożkiewicz

Lublin University of Technology

Poland

Reviewer 2 Report

This article diagnoses the dynamic response of rolling bearings according to radial internal clearance values.  However, there are some issues in the article that need to be corrected:

1. The article lacks a flowchart illustrating the overall application method. The chart should outline the steps of the proposed method and the innovative parts in a generalized way.

2. The number of literature reviews is insufficient. Some articles are too old, more attention should be paid to the recent development of fault diagnosis.

3. There are some problems with the writing style and framework of the thesis. The method should be introduced before the experimental part. The explanation of the proposed method is limited in length and content, which is not conducive for the author to expound on their innovations.

well

Author Response

Lublin, 17.06.2023

Response to Reviewer no. 2

Dear Reviewer,

We appreciate the constructive suggestions enclosed in your review. All the authors of this manuscript are grateful for Reviewer’s encouraging approach, as well as constructive criticism that is intended to improve our work. We carefully considered the comments
and herein we explain how we revised the manuscript based on your recommendations.

It is our belief that the manuscript is substantially improved after making the suggested edits. We want to express our appreciation for taking the time and effort necessary to provide guidance.

Yours sincerely

Bartłomiej Ambrożkiewicz

Lublin University of Technology

Poland

Reviewer 3 Report

It was hard to find any contribution to this paper. But, if the author could publish the dataset, that should be great for other researchers.

Some questions should be considered by the authors.

1. What is the contribution of this paper, the VMD and the classification are all general methods.

2, Why do you carry out experiments using different radial internal clearances? What kind of conclusion do you get from the results?

3, I suggest the author could conduct more signal analysis for all different experiments with different clearances. You could follow the method proposed in https://www.sciencedirect.com/science/article/pii/S0888327017300754, (Research on variational mode decomposition in rolling bearings fault diagnosis of the multistage centrifugal pump

) which also used VMD.

4, I suggest the authors could consider the deep neural network for the classification task.

Author Response

Lublin, 17.06.2023

Response to Reviewer no. 3

Dear Reviewer,

We appreciate the constructive suggestions enclosed in your review. All the authors of this manuscript are grateful for Reviewer’s encouraging approach, as well as constructive criticism that is intended to improve our work. We carefully considered the comments
and herein we explain how we revised the manuscript based on your recommendations.

It is our belief that the manuscript is substantially improved after making the suggested edits. We want to express our appreciation for taking the time and effort necessary to provide guidance.

Yours sincerely

Bartłomiej Ambrożkiewicz

Lublin University of Technology

Poland

Round 2

Reviewer 1 Report

This is ok.

This is ok.

Reviewer 3 Report

The revision is ok for me.